# INITIALIZING THE LAYER-WISE LEARNING RATE

## ABSTRACT

Weight initialization schemes have been devised with heavy emphasis in the initial training dynamics, assuming the optimizer automatically handles appropriate step sizes in prolonged training. The optimizer typically calculates the step sizes using a single, global learning rate across all parameters, focusing exclusively on the (exponentially averaged) in-training time gradient. Motivated from hierarchical structure inherent in deep networks, this work explores assigning non-adaptive layer-wise learning rates based on the differences in gradient magnitude at initialization as a practical and effective optimization strategy. The gradient magnitude used to preset the layer-wise learning rates is measured at fan-in initialization, as stable activation variance is considered a desirable property during training, and so is assumed to largely hold true in prolonged training. Experiments on convolutional and transformer architectures show the proposed layer-wise learning rate can improve training stability and convergence in image classification and autoregressive language modeling

## 1 INTRODUCTION

Gradient descent has become the de-facto method for training deep neural networks, with various optimizers (e.g., SGD, RMSProp, and Adam) differing in how they determine the step size for updating parameters from the calculated gradients. The weight update process via gradient descent can be broadly divided into two stages: first, the optimizer determines the per-parameter step sizes, and second, they are scaled by the learning rate, which is a hyperparameter. This two-stage process reflects the empirical and theoretical nature of modern neural network training, where the hyperparameter is empirically determined and theoretically sound techniques are employed by the optimizer to ensure wide generalizability and minimize unnecessary hyperparameter tuning.

While adjusting the layer-wise learning rate was initially explored as a means to improve neural network training (LeCun et al., 2002), adaptive step size scaling proved to be widely successful in improving neural network training and has become the default gradient descent variation when training in difficult settings (Duchi et al., 2011; Kingma & Ba, 2015). SGD can show strong performance in low data regimes but can be more difficult to converge, making it an unattractive option when training complicated networks on large datasets. Appropriate weight initialization is also important for successful training (LeCun et al., 2002; Glorot & Bengio, 2010), and both adaptive step-size scaling and intricate weight initialization schemes have been extensively explored. However, it is unclear to what extent commonly used theoretical assumptions carry on to empirical optimizer performance (Tran et al., 2024) and the architectural implications on training continue to be an active area of research (Kunstner et al., 2023; Zhang et al., 2024a).

Modern architectures are characterized by the use of multiple linear and nonlinear operations before the final output, and efficient hierarchical learning is an important factor for the success of neural networks (Chen et al., 2020; Abbe et al., 2021; Allen-Zhu & Li, 2023). Layer depth is known to result in different convergence speeds for convolutional architectures (Zeiler & Fergus, 2014; Yosinski et al., 2014), and transfer learning and finetuning settings have made wide use of explicit layer-wise learning rates to train selected layers for improved performance (Donahue et al., 2014; Sharif Razavian et al., 2014; Girshick et al., 2014; Kumar et al., 2022; Guo et al., 2019; Lee et al., 2019; Ro & Choi, 2021; Howard & Ruder, 2018; Bao et al., 2022; Lee et al., 2023). The justification is that when finetuning on a smaller dataset, further training of general-purpose, low-level layers that are close to the input is likely to overfit and worsen performance compared to only training the specialized, high-level layers that are close to the output. Given that intermediate feature characteristics are in

many respects hard-coded in architecture, it is reasonable to expect that explicit layer-wise learning rates can also be beneficial when training from scratch.

This work focuses on the simplest version that assigns static learning rates across the entire training duration, as architectural characteristics such as input-output dimension, nonlinear activation functions, and layer-wise connectivity are static and do not change during training. From the observation that gradient magnitude at initialization correlates with commonly accepted layer-wise convergence in convolutional networks, a scheme that assigns relative layer-wise learning rates with respect to the gradient magnitude at initialization can be devised by interpreting the gradient magnitude at initialization as a measure of architecture-induced convergence bias that persists during prolonged training.

The gradient magnitude used as a basis to adjust the layer-wise learning rate is measured when layer weights are initialized to preserve the activation variance, as stable activation variance is widely considered a desirable property (Ioffe & Szegedy, 2015; Brock et al., 2021; Roberts et al., 2022). After measuring the gradient magnitude, the relative layer-wise learning rates are initialized to be opposite the layer-wise gradient magnitude. By regularizing the architecture-induced convergence bias, it can be considered as a preventative measure to the exploding gradient problem by extrapolating from statistics at initialization, making it of conceptually different approach from methods that aim to explicitly normalize or clip the gradient during training (Zhang et al., 2018b; 2020).

Experiments show improved performance and training stability on ImageNet-1k classification and 124M GPT-2 autoregressive language modeling. The improved convergence is represented in the ability to handle higher learning rates and achieve lower training loss, which is an intriguing phenomenon given the simplicity of the method compared to techniques that involve continuous tracking of per-parameter gradient statistics. Inspecting the actual layer-wise learning rates values shows that low learning rates are assigned to low-level layers and high learning rates to deeper, high-level layers, but with additional subtleties and details as they are assigned with per-layer granularity. Altogether, the empirical results suggest that adjusting the layer-wise learning rate is a simple yet effective method for improving convergence, requiring negligible in-training time memory and computation overhead while being compatible with contemporary adaptive methods.

## 2 RELATED WORK

**Weight initialization.** Weight initialization schemes were proposed as a solution to the exploding/vanishing activation/gradient problem in vanilla feed-forward networks (Glorot & Bengio, 2010). He et al. (2015) additionally incorporates the effect of subsequent ReLU into weight initialization, and for residual connections, Zhang et al. (2019b) and De & Smith (2020) suggest downscaling the branch of residual blocks due to the increasing activation variance with depth. Formal explanations have been developed on the trainability of deep networks using mean field theory (Poole et al., 2016; Schoenholz et al., 2017; Xiao et al., 2018), and orthogonal weight initialization demonstrated the ability to convolutional networks of extreme depths, albeit it does not necessarily outperform the shallower counterparts (Saxe et al., 2014; Xiao et al., 2018).

In practice, basic initialization schemes are widely used in conjunction with normalization layers (Ioffe & Szegedy, 2015; Ba et al., 2016; Zhang & Sennrich, 2019) to train deep networks. Some explanations for the success of normalization methods include prevention of activation explosion (Bjorck et al., 2018), smoothening of the optimization landscape (Santurkar et al., 2018), and prevention of rank collapse (Daneshmand et al., 2020). The rise of recent transformer networks spurred its dedicated initialization analysis (Xiong et al., 2020; Liu et al., 2020; Huang et al., 2020; Zhang et al., 2019a), while other works have argued for the need of architecture agnostic initialization schemes (Dauphin & Schoenholz, 2019; Zhu et al., 2021).

**Layer-wise optimization.** Many optimizers that involve layer-wise elements have been proposed, such as directly using the layer-wise gradient history (Singh et al., 2015) or incorporating additional mechanisms such as gradient normalization and weight norm scaling (You et al., 2017; Yu et al., 2017; Zhou et al., 2019; You et al., 2020; Heo et al., 2021; Bernstein et al., 2020; Liu et al., 2021a). Another direction previous works indirectly modified the layer-wise learning rate is through the use of scale factors, most commonly included at the residual branches (Karras et al., 2018; Hayou et al., 2021; Liu et al., 2020; Touvron et al., 2021; Noci et al., 2022). Such scale factors can affect both the

gradient size and effective weight deviation of the layer, resulting in different effects depending on whether the optimizer scales the step size depending on the gradient magnitude or variance (Balles & Hennig, 2018). This work differs in that the architecture remains unmodified and instead modifies the layer-wise learning rates at initialization, so individual layers are updated with different step sizes even if the gradient or weight statistics at training time were to be identical.

Efficient methods to adapt the Hessian into the optimizer have been explored as a promising candidate to improve training (Martens & Grosse, 2015; Gupta et al., 2018; Yao et al., 2021; Liu et al., 2024) and meta-learning settings made use of layer-wise hyper gradient to improve performance and efficiency (Antoniou et al., 2019; Baik et al., 2020; Tang et al., 2021; Chen et al., 2023b). Other contexts for exploring per-layer learning rates include curriculum learning (Croitoru et al., 2024), efficient hyperparameter transfer (Yang et al., 2021), heavy-tailed self-regularization theory (Zhou et al., 2023) and for reducing the memory footprint of AdamW (Zhang et al., 2024b).

## 3 BACKGROUND

Gradient descent is an algorithm derived from the principle that for a smooth and differentiable loss function $\mathcal{L}$ defined with respect to parameters $\theta$, moving in the direction of the negative gradient of $\mathcal{L}$ at $\theta$ will decrease the function's value most rapidly. This negative gradient direction formally refers to $-\nabla_\theta \mathcal{L}(\theta)$.

Many loss functions are defined in convex or pseudoconvex to facilitate convergence to a minimum when maximizing the negative likelihood of the data distribution. To optimize the parameters towards the minimum, the parameters $\theta^k$ are updated at each iteration step $k$ by moving against the gradient of the loss function as follows:

$$\theta^{k+1} = \theta^k - \gamma \nabla_{\theta^k} \mathcal{L}(\theta^k), \tag{1}$$

where $\gamma \in \mathbb{R}_+$ is the learning rate controlling the step size for each update. Typically, well-known optimizers process the gradient before it is multiplied by the learning rate, as shown below:

$$\theta^{k+1} = \theta^k - \gamma O(\nabla_{\theta^k} \mathcal{L}(\theta^k)), \tag{2}$$

where $O(\cdot)$ represents the optimizer-specific preprocessing performed to calculate the step size. For example, for Stochastic Gradient Descent (SGD) with momentum $m$, the process can be represented as follows:

$$O(\nabla_{\theta^k} \mathcal{L}(\theta^k)) = \mathbf{v}^k; \quad \mathbf{v}^k = m\mathbf{v}^{k-1} + \nabla_{\theta^k} \mathcal{L}(\theta^k). \tag{3}$$

Here, the initial $\mathbf{v}^0$ is initialized to zero vector.

## 4 LAYER-WISE LEARNING RATES

The parameters $\theta$ in a deep neural network can be naturally grouped into different parameter blocks according to their layer-wise structure. Let $\theta = \{\omega_1, \cdots, \omega_L\}$, where $L$ is the total number of layers. In this work, we consider layer-wise learning rates defined as multiplying the learning rate by relative learning rate values $\eta_l$ for each parameter block $\omega_l$, corresponding to the parameters of a $l$-th layer as follows:

$$\omega_l^{k+1} = \omega_l^k - \gamma \eta_l O(\nabla_{\omega_l^k} \mathcal{L}(\omega_l^k)). \tag{4}$$

Assigning a unique layer-wise learning rate for each $l$-th layer $\eta_l$, which does not change throughout training, incurs negligible overhead, and is supported by popular frameworks.

### 4.1 MOTIVATION

Figure 1 shows the averaged per-parameter additional step size scaling performed by AdamW (Loshchilov & Hutter, 2019) and Lars (You et al., 2017) compared to SGD momentum, and Lamb compared to AdamW (You et al., 2020). The step size due to the second moment in AdamW shows a clear layer-wise pattern that is consistent across the entire training duration, showing that there is a clear per-layer trend in the parameter-wise gradient history. Lars and Lamb multiplies the step size

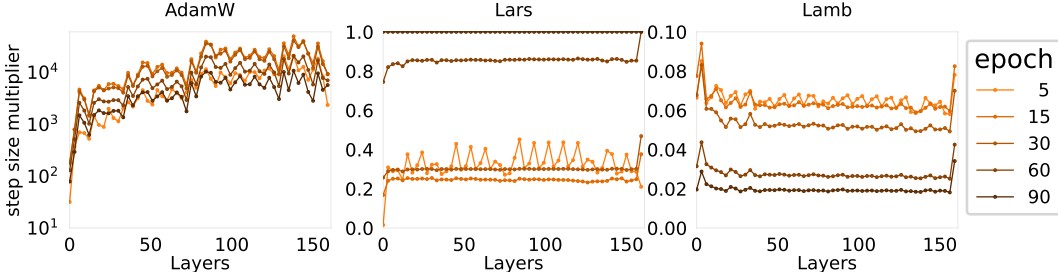

Figure 1: Additional per parameter step size multiplier compared to SGD due to AdamW second moment (left) and Lars (middle), and Lamb's step size multiplier compared to AdamW(right) when training ResNet-50 for 90 epochs. Only showing convolutional/linear layers for visibility. For LARS, trust coefficient of 0.001 is used and the step size multiplier is clipped to 1. Lamb's additional step size multiplier is only performed to the conv/linear layers while batch norm layers are not modified and are fixed to 1.0, creating a large discrepancy.

by a combination of both a layer-wise weight norm and gradient normalization, and its additional step size scaling performed upon SGD is surprisingly consistent across layers despite the layer-wise naming might suggest.

Adaptive methods come at the cost of extra memory used to store the per-parameter gradient history and treat all parameters in a unified manner. Per-layer analysis can be a convenient middle ground for interpreting and manipulating neural networks as it has a lower dimension to search compared to per-parameter analysis (Bengio et al., 2006; Baldock et al., 2021; Pan et al., 2023). Given the clear layer-wise trend of adaptive methods when training from scratch, it is natural to investigate the simplest version that assigns static learning rates across the entire training duration.

**Generalization.** Using high learning rates has been associated with beneficial regularizing effects at the cost of potential difficulty in convergence (Goyal et al., 2017; Li et al., 2019; Lu et al., 2024), but is not immediately clear how it can be leveraged to a layer-wise level. Heavier tails in the empirical spectral density of individual layers have been linked with better generalization Mahoney & Martin (2019), and Zhou et al. (2023) fits a power law distribution on the layer-wise empirical spectral density in order to extract a layer-wise generalization measure and adjusts the layer-wise learning rate accordingly. The assigned learning rates show a low to high trend throughout most of the training.

**Balanced training.** Balancing convergence of the weights to ensure stable training is a common motivation for modifying the learning rate, and convergence has been measured by tracking gradient, weight, neuron output or loss (Raghu et al., 2017; Bragagnolo et al., 2022; Liang et al., 2022; Du et al., 2022). The various methods to measure convergence show the difficulty in determining if the current weight is close to the final value before training has ended since weights considered to have converged may require further training as the remaining weights are modified (Bragagnolo et al., 2022). Preemptive freezing of selected weights may also make the network brittle to weight deviations of those frozen prematurely, resulting in reduced generalization. While the use of in-training time statistics is extensively explored in various literature, it may not appropriately incorporate more global aspects of the training dynamics that are not apparent from the immediate gradient history (Micaelli & Storkey, 2021), and leveraging ahead-of-time information can be an effective complementary method.

**Preserving activation.** The scale of activations at initialization can be modified by manipulating the scale of layer weights, and preserving activation variance has been a major motivation for weight initialization techniques. He et al. (2015) additionally scales the weight initialization by $\sqrt{2}$ in consideration of the ReLU non-linearity, which, in view of the linear layer, means increasing the activation variance before it passes through the ReLU. In the context of initialization only, activation variance could also be preserved by increasing the ReLU slope, essentially introducing an additional multiplicative constant instead of just scaling the weight. Similarly, introducing an additional $1/\sqrt{2}$ multiplicative constant when summing the residual block output can prevent exploding variance on

---

**Algorithm 1:** Layer-wise learning rate initialization of neural networks

**Input:** Model, layer weights $\theta = \{\omega_1, ..., \omega_L\}$ and numbers of parameters $\{N_1, ..., N_L\}$
**Output:** Relative layer-wise learning rates $\eta_l$

1: Set $G_l^0 = 0, \; l \in \{1, ..., L\}$
2: Initialize conv/linear layers from $\mathcal{N}(0, 1/f_{in})$, scale layers to **1**, bias layers to **0**
3: **For** $t \leftarrow 1$ **to** $T$
4:      Sample minibatch from training set
5:      $\boldsymbol{g}_\theta^t \leftarrow \nabla_\theta \mathcal{L}^t(\theta)$
6:      **For** $l \leftarrow 1$ **to** $L$
7:          $G_l^t \leftarrow G_l^{t-1} + \frac{1}{N_l} \sum_{i \in \omega_l} |\boldsymbol{g}_i^t|$
8: **For** $l \leftarrow 1$ **to** $L$
9:      $\tilde{\eta}_l \leftarrow \frac{1}{\sqrt{G_l^T}}$
10: $N_{sum} = \sum_{l=1}^{L} N_l$
11: $\tilde{\eta}_{sum} = \frac{1}{N_{sum}} \sum_{l=1}^{L} \tilde{\eta}_l N_l$
12: **For** $l \leftarrow 1$ **to** $L$
13:      $\eta_l \leftarrow \frac{\tilde{\eta}_l}{\tilde{\eta}_{sum}}$

---

ResNets (Balduzzi et al., 2017). In practice, such additional multiplicative factors are not widely used compared to explicit normalization techniques such as batch normalization.

## 4.2 REGULARIZING ARCHITECTURE-INDUCED CONVERGENCE BIAS

This work takes on the view that architectural characteristics are reflected in the gradient as it is back-propagated through the network and that the difference in gradient magnitude when layers are initialized to preserve its immediate activation variance represents the layer-wise convergence bias due to architectural factors. After measuring the layer-wise gradient magnitude at initialization, the relative learning rates can be adjusted to counteract the differences in convergence bias. It effectively extrapolates from statistics at initialization when the network has not undergone any training and under the assumption that the activation scale would not deviate far from initialization during prolonged training.

Algorithm 1 outlines the proposed layer-wise learning rate initialization. It starts by initializing all convolution/linear weights from random numbers sampled from $N(0, 1/f_{in})$, where $f_{in}$ is the fan-in number. It is a basic initialization scheme that preserves the activation variance if the weights and gradients are i.i.d. (Glorot & Bengio, 2010). As standard, scale layers are initialized to 1 and bias layers to 0. The class token, position embedding, and relative position layers in transformer architectures are treated as biased and initialized to 0. The weight-tied embedding/head layer of language models is initialized with a standard deviation of $(1 + \sqrt{1/f_{in}})/2$ as a middle ground of ensuring unit normalized input to the network and the fan-in initialization of the head layer.

Next, the layer-wise gradient magnitude is collected for $T$ iterations while the model weights are fixed at initialization. The gradient magnitude is collected per parameter as the learning rate is applied on a per-parameter basis, and the relative layer-wise learning rate is adjusted inversely proportional to the square root of the gradient magnitude and then normalized. As the average per-parameter gradient magnitude per layer is used, the method is not sensitive to the value of $T$. For ResNet-50, using a $T$ of 1 iteration instead of 5004 results in a max 11.02%, average 2.66% deviation in the assigned layer-wise learning rates, which would not significantly impact the network performance compared to other random factors. 5004 is the number of iterations for 1 epoch when training with a batch size of 256. We would like to note that the square root in line 9 is necessary for the proposed learning rate to be beneficial (see Table 2). The intuition is that architectural convergence bias is partially represented in the in-training time gradient, thus requiring a weaker scaling.

Table 1: Final top-1 validation accuracy (↑) for ImageNet-1k classification. Trained without gradient clipping and label smoothing. Average and standard deviation of 3 runs reported.

| Model | #Params | Optimizer | Data Augmentation | Epochs | Learning Rate | |
|---|---|---|---|---|---|---|
| | | | | | Single | Layer-wise |
| ResNet-50 | 25.56M | SGD | basic | 90 | 76.91±.12 | 77.03±.09 |
| | | | | 200 | 77.24±.08 | **78.08±.06** |
| | | AdamW | basic | 200 | 76.57±.08 | **77.01±.06** |
| | | | strong | 300 | 78.25±.20 | **78.75±.08** |
| ViT-S/16 | 22.05M | SGD | basic | 300 | 69.01±.32 | **71.84±.51** |
| | | AdamW | basic | 300 | 75.46±.36 | 75.42±.17 |
| | | | strong | 300 | 77.39±.43 | **78.10±.17** |
| Swin-T | 28.29M | AdamW | strong | 300 | 79.89±.15 | 79.95±.09 |
| ConvNeXt-T | 28.59M | AdamW | strong | 300 | 80.26±.15 | 80.43±.03 |

## 5 EVALUATION

The proposed layer-wise learning rate scheme is evaluated on ImageNet-1k and CIFAR-100 image classification and on autoregressive language modeling. The experiments demonstrate that appropriate layer-wise learning rates can improve convergence on both various optimizers and architecture archetypes despite the simplicity and non-existent overhead of using static relative layer-wise learning rates over the entire training run. We also inspect the assigned layer-wise learning rates, which show a trend to increase according to depth along with other intricacies. Ablation studies on the design choices of the layer-wise learning rate assigning algorithm are performed in small-scale dataset overfitting experiments in autoregressive language modeling.

### 5.1 IMAGENET-1K

**Experiment settings.**  We perform experiments on ResNet-50 and ViT-S/16 when training from scratch on basic Inception-style preprocessing and strong data augmentation. We would like to note the aim is not to achieve SOTA results but to evaluate on generic settings. For hyperparameters such as initial learning rate, effective batch size, and stochastic depth, we mostly rely on the values reported in Chen et al. (2023a). When training ResNet-50 with SGD, we use a batch size of 256 and weight decay of 1e-4 similar to Goyal et al. (2017). For strong data augmentation, we use a combination of RandAugment with layer 2 and magnitude 10 (Cubuk et al., 2020) and Mixup with strength 0.5 (Zhang et al., 2018a). More detailed settings are reported in the Appendix.

**Results.**  Table 1 show that the proposed layer-wise learning rates can improve training, even for SGD which is known to have strong performance when training convolutional networks under simple data preprocessing pipelines. The accuracy increase is particularly notable when training for 200 epochs, improving by 0.96% from 77.24% to 78.19%. This has significant implications because it is a setting where SGD is known to already achieve higher accuracy compared to adaptive optimizers, which is often attributed to the tendency of adaptive optimizers to overfit more easily. The fact that the performance of SGD can be further improved by a technique indicates the existence of an unexplored direction for improving optimization. This is further emphasized by the fact that the proposed layer-wise learning rates are also beneficial with AdamW, improving accuracy by 0.50% from 78.25% to 78.75% when training for longer epochs under stronger data augmentation. The train loss curve in Figure 2 shows that the proposed layer-wise learning rate achieves lower train loss in the initial stage where the learning rate is high, suggesting it can better handle higher learning rates. It also suggests the capability of resnets could have been underrepresented when training with a single learning rate.

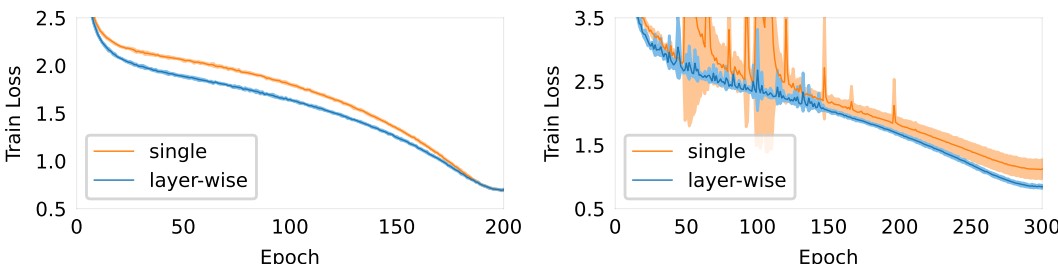

Figure 2: ResNet-50 (left) and ViT-S/16 (right) train loss (↓) when trained with basic data augmentation and SGD on ImageNet-1k classification.

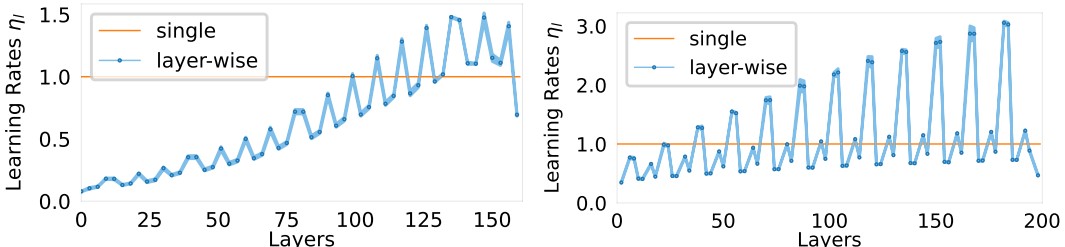

Figure 3: Assigned learning rates to ResNet-50 (left) and ViT/S-16 (right). We only show convolution/linear layers for visibility, which comprise 99.8% of total parameter count.

The improved convergence of the proposed layer-wise learning rate is more clearly visible when training vision transformers with SGD, which is a challenging task where previous works applied weight reparameterization or exotic optimization techniques (Zhai et al., 2023; Chen et al., 2022). The train loss curves in Figure 2 show significantly lower and stable training loss throughout the entire training run and is reflected in an accuracy improvement of 2.84%. Interestingly, there does not seem to be a clear improvement when training with AdamW under basic augmentation, which contrasts with ResNet-50, where the layer-wise learning rate was also beneficial for AdamW. Training transformer architectures are known to be prone to overfitting compared to convolutional architectures when training on image data, possibly diluting the effect of layer-wise learning rates in this setting. When training with stronger data augmentations, we find that the layer-wise learning rate achieves more consistent results, resulting in visible accuracy improvements over multiple runs. Experiments on more sophisticated architectures such as ConvNeXt-T (Liu et al., 2022) and Swin-T (Liu et al., 2021b) demonstrate that the layer-wise learning rates remain competitive despite the effort involved in modifying the architecture.

**Inspecting assigned learning rates.** Figure 3 shows the assigned layer-wise learning rates for ResNet-50, which has a 3-4-6-3 bottleneck block configuration. The layer-wise learning rates show a clear depth-wise trend except for the last head layer, which is assigned a relatively lower learning rate. Since numerous layers exist in a typical network, especially when considering scale and bias as separate layers, only the assigned learning rates to the convolution and linear layers are shown as they comprise most of the model parameter count. ResNet-50 consists of multiple residual bottleneck blocks, which individually have 3 convolution layers, and the repetition of bottleneck blocks is demonstrated through the pattern of a low learning rate for the first two convolution layers and a higher learning rate for the third convolution layer in the bottleneck block. We find scale and bias layer learning rates are generally similar to the directly preceding linear layer, although the bias layer has a tendency to be assigned higher learning rates, especially for the intermediate stage that consists of 6 consequent bottleneck blocks. The bias layer of the third convolution layer in the 6th bottleneck block is assigned 2.13, which is significantly larger than the 1.39 assigned to the preceding convolution layer.

ViT/S-16 consists of 12 blocks, where each block is comprised of a self-attention block that has 4 linear layers and a multilayer perceptron block that has 2 linear layers. A similar depth-wise trend in the assigned learning rates can be observed in Figure 3, although it is much weaker than ResNet-50.

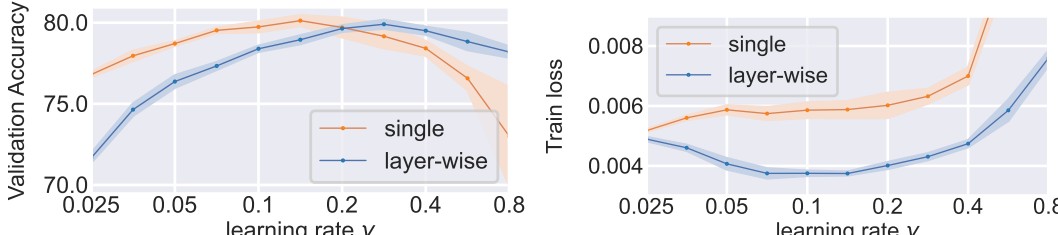

Figure 4: Validation accuracy (left ↑) and train loss (right ↓) when training ResNet-50 on CIFAR-100 for 200 epochs. Results of 5 runs.

The spikes in assigned learning rates correspond to the query and key layers of the self attention block, which show that their gradient is very low compared to other layers at initialization. Similar findings have been reported in Noci et al. (2022), where it only analyzed the gradient discrepancy in the attention block and proposed an inverse temperature scaling. Our approach adjusts the learning rate and is applied equally to all layers in a network. The first convolution layer in the stem patch embedding is assigned a low learning rate of 0.347, while the class token layer and the position embedding layer are assigned a lower 0.012 and 0.17. Figures for ConvNeXt-T and Swin-T are provided in the appendix.

## 5.2 CIFAR-100

**Experiment settings.**   We also evaluate the effect of layer-wise learning rates on CIFAR-100 classification when training from scratch on ResNet-50 with SGD. We sweep over learning rates when trained for 200 epochs with 1 epoch warmup under the standard data preprocessing. We use a batch size of 256, weight decay of 5e-4 and perform cosine learning rate decay schedule. The smaller dataset compared to ImageNet-1k can make it more sensitive to the initial weights, so both single and layer-wise learning rates start training from the same fan-in weight initialization scheme.

**Results.**   Figure 4 shows that the single learning rate has higher accuracy in lower global learning rates, while layer-wise learning rates have an advantage when the global learning rate is higher. With regard to accuracy, it is unclear which is better as both achieve similar accuracy but with different optimal hyperparameters. This demonstrates the nuances required when evaluating performance from small scale datasets, as we previously demonstrated layer-wise learning rates is beneficial for ImageNet-1k under hyperparameters likely to be tuned for single learning rate. However, it can be observed in Figure 4 that layer-wise learning rate consistently achieves lower train loss on high learning rates. It suggests it may be possible to extrapolate performance from small scale experiments, but perhaps not necessarily from only looking at the highest possible accuracy, and Wortsman et al. (2024) suggest that techniques that stabilize training in small-scale experiments in high learning rate regimes can also be carried on to larger scales.

## 5.3 AUTOREGRESSIVE LANGUAGE MODELING

**Experiment settings.**   Autoregressive language modeling is used to pretrain large models on large corpus of data, and demonstrates an interesting challenge where both the model and dataset can become very large, making iterating over the dataset over multiple epochs infeasible. Due to the costs involved in large scale training, we evaluate on the 124M GPT-2 model. We use a variant that does not have bias on the linear and layer norm layers, and use $\beta_1$, $\beta_2$ of 0.9, 0.95 for AdamW, 0.95, 0.98 for Lion (Chen et al., 2023a) and 0.965, 0.99 for Sophia (Liu et al., 2024). The assigned layer-wise learning rates are shown in Figure 5 and show similar pattern to ViT-S/16, except for the large learning rates of 2.74 and 0.66 assigned to the embedding/head and positional encoding layers.

We evaluate two settings. The first setting is where a small subset of data is repeatedly iterated over multiple epochs, similar to typical image settings, which we argue can still be a reasonable proxy for long-term training behavior. The second is on 9.89B token training, where each minibatch of data is seen only once. All experiments start training the same weight initialization that samples weights from $\mathcal{N}(0, 0.02^2)$ for the linear layers, except for the weights of residual branches which

Table 2: Train loss ($\downarrow$) of small scale overfitting experiments with AdamW optimizer on 124M next token prediction. Ablation study performed on fan-in and fan-out weight initialization when calculating the layer-wise learning rates. * denotes when scaling without performing the square root on the layer-wise gradient magnitude.

| Learning rate $\gamma$ | 7.5e-5 | 1.5e-4 | 3e-4 | 6e-4 | 1.2e-3 | 2.4e-3 | 4.8e-3 | 9.6e-3 |
|---|---|---|---|---|---|---|---|---|
| Single | 3.666 | 2.672 | **1.430** | 2.095 | 2.334 | 6.965 | – | – |
| Layer-wise ($f_{in}$) | 4.169 | 3.349 | 2.114 | 0.974 | **0.560** | 1.163 | – | – |
| Layer-wise ($f_{out}$) | 4.099 | 3.260 | 2.139 | 1.299 | **0.808** | 1.675 | – | – |
| Layer-wise ($f_{in}$*) | – | – | 3.466 | 2.775 | 1.946 | **1.532** | 1.573 | 1.719 |

Table 3: Train loss ($\downarrow$) of small scale overfitting experiments with Lion optimizer on 124M next token prediction.

| Learning rate $\gamma$ | 1.5e-5 | 3e-5 | 6e-5 | 1.2e-4 | 2.4e-4 | 4.8e-4 |
|---|---|---|---|---|---|---|
| Single | 2.966 | 1.833 | **0.704** | 1.312 | 2.559 | 7.477 |
| Layer-wise ($f_{in}$) | 3.387 | 1.765 | 0.542 | 0.270 | **0.179** | 0.346 |

are sampled from $\mathcal{N}(0, 0.02^2/(2*B))$. $B$ is the number of attention blocks, which for 124M GPT-2 is 12.

**Overfitting on small dataset.** Full training can be costly in terms of both energy and time, and a common strategy is to extrapolate from smaller-scale experiments that can be performed in a shorter time frame. While the setting of interest views each minibatch of data once, the large corpus of data makes it likely that the underlying data characteristics are shared and reflected across many minibatch of data, even if the exact sequence of tokens is not identical. A setting where a smaller subset of the dataset is repeatedly iterated over can be an efficient method to check if it can learn the characteristics implicitly shared across many minibatches, at the drawback of not explicitly verifying it on the large, diverse full dataset.

Table 2 shows the final train loss when sweeping over various global learning rates on AdamW. We repeatedly iterate over 20 minibatch of data for 2000 iterations with a constant learning rate schedule and measure the average train loss of the respective minibatchs. The layer-wise scheme achieves lower train loss and can handle higher global learning rates, demonstrating that appropriate layer-wise rates can improve convergability. We also perform an ablation study on the design choices of Algorithm 1 and verify that measuring the gradient at fan-in initialization and scaling inversely proportional to the square root of the gradient magnitude achieves the lowest train loss. We also find that it improves train loss in the small scale experiments on recently proposed optimizers such as Lion in Table 3 and also on Adam-mini and Sophia in Table 8 and Table 9 in the Appendix.

Table 4: Final validation loss ($\downarrow$) on 9.89B token training, 124M GPT2 model.

| Learning Rate | AdamW | Lion | Adam-mini | Sophia |
|---|---|---|---|---|
| Single | **3.3344** | 3.2888 | **3.3707** | 3.3523 |
| Layer-wise | 3.3530 | **3.2851** | 3.3844 | **3.3153** |

**9.89B token training.** Next we evaluate on a setting where each minibatch is only seen once on the Fineweb dataset (Penedo et al., 2024). We use a learning rate that is double the optimal value found in the previous small dataset overfitting experiment, as a cosine learning rate schedule is used. This is a setting where we expect fewer improvements, as the advantage of layer-wise learning rates is in prolonged training where the balanced training of layers allows the network to maintain its convergability. The single learning rate scheme performs high step sizes to layers with high gradients, quickly adapting to individual examples but potentially at the cost of further learning capacity in later stages, as demonstrated in the previous small-scale overfitting experiments.

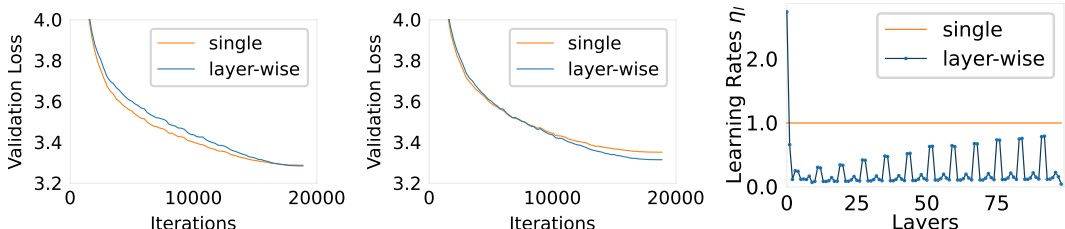

Figure 5: Validation loss (↓) when training 124M GPT2 model on 9.89B tokens with Lion (left) and Sophia (middle). The assigned layer-wise learning rates to 124M GPT2 model (right).

Table 4 shows the final validation loss for 4 optimizers, and shows that there is indeed a less remarkable improvement in the final validation loss. In fact, there is a slight degradation for AdamW and Adam-mini, which considers its similarity to AdamW as a strength. The layer-wise scheme is beneficial to more modern optimizers such as Lion and Sophia, with the benefit with Sophia being particularly noticeable. The training curves Figure 5 shows that the layer-wise learning rate initially lags behind but later catches up in the later stages. Given that this setting is comparatively much shorter compared to typical image settings, we expect there will be a more significant improvement when training for longer iterations: a single epoch of 9.89B tokens training is 18,865 iterations on a 124M parameter model, which is significantly lower than the 93,825 iterations taken to train ImageNet-1k on a smaller 22M ViT-S/16 model.

Table 5: Train loss (↓) of small scale overfitting experiments using Lion optimizer on 774M model. $\beta$ indicates with residual downscaling as a constant multiplicative scaler that is not trained.

| Learning rate $\gamma$ | 3.75e-5 | 7.5e-5 | 1.5e-5 | 3e-5 | 6e-5 | 1.2e-4 | 2.4e-4 |
|---|---|---|---|---|---|---|---|
| Single | 2.647 | 0.840 | 0.316 | 0.092 | 0.301 | 4.952 | - |
| Layer-wise ($f_{in}$) | 2.528 | 0.452 | 0.105 | 0.241 | 0.488 | 7.174 | - |
| Single-$\beta$ | - | 2.513 | 0.367 | 0.059 | 0.156 | 6.403 | 7.537 |
| Layer-wise-$\beta$ ($f_{in}$) | - | 0.748 | 0.034 | **0.019** | 0.026 | 0.056 | 7.492 |

**Scaling with depth.** Downweighting the residual branches $h(x) = x + \beta f(x)$ is a prevalent modification investigated in various initialization works (De & Smith, 2020; Hayou et al., 2021; Noci et al., 2022), and downscaling by $\beta = O(1/\sqrt{depth}) < 1$ is suggested to prevent rank collapse at initialization when scaling with depth. We experiment using Lion optimizer on a 774M model which has 36 attention blocks using a $\beta$ of $1/\sqrt{72}$, and compare between the single and layer-wise learning rate with and without the multiplicative residual downscaling. Table 5 shows that without the residual downscaling the single learning rate achieves slightly lower train loss in small scale experiments, but with $\beta$ downscaling the layer-wise scheme has both the lowest train loss and wider range of learning rates that achieve low training loss. It shows that the layer-wise learning rate scheme is scalable with depth and can benefit further from advances in initialization schemes.

## 6 DISCUSSION

The proposed layer-wise learning rate adjusting scheme is conceptually simple and incurs very little overhead, and empirical results demonstrate that it can substantially improve convergence when used in conjunction with widely used optimizers. However, it is based on the assumption that extrapolating from statistics at initialization can be used to model long-term behaviors that are difficult to measure exclusively from in-training time gradients. While there are reports that extremely large activations exist in fully trained large models (Sun et al., 2024), such activations are relatively scarce and most activations are reported to be stable. We believe that the simplicity and efficacy of the method make it a promising direction for analyzing and improving modern neural network training.

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

## A APPENDIX

**ImageNet-1k setting details.**   The experiments were run using pytorch (Paszke et al., 2019) and timm (Wightman, 2019) for transformer model architectures and data preprocessing. Pytorch automatic mixed precision training with bfloat16 was used, which we found to significantly improve ViT-S/16 performance when training with AdamW and basic augmentation compared to using float16 with loss scaling. We do not perform gradient clipping or label smoothing but include strong data augmentations to demonstrate the effectiveness of appropriate layer-wise learning rates under more difficult training settings.

All experiments were trained using a cosine decay learning rate schedule with a warmup, image resolution of $224 \times 224$, and default momentum/beta hyperparameters for SGD and AdamW. Gradient accumulation was performed to simulate larger effective batch sizes. When adjusting the layer-wise learning rates, the gradient was collected over an epoch, which corresponds to a $T$ of 312∼5004, depending on the batch size. All reported ImageNet-1k experiments took ∼150 GPU days of training.

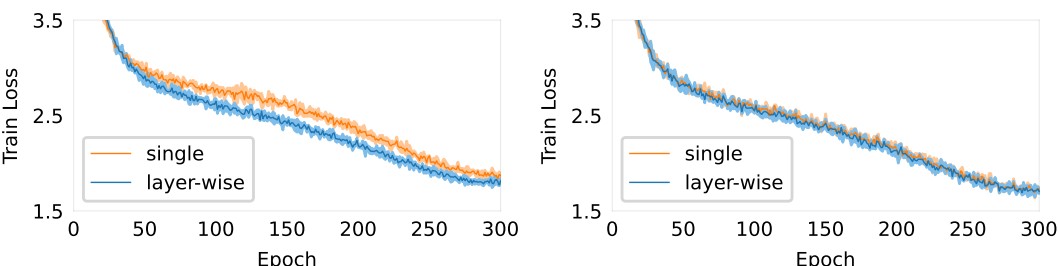

Figure 6: Swin-T (left) and ConvNeXt-T (right) train loss when trained with strong data augmentation and AdamW on ImageNet-1k classification.

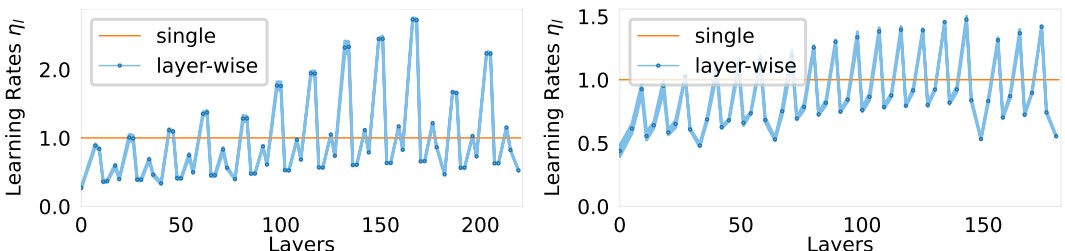

Figure 7: Assigned learning rates to Swin-T (left) and ConvNeXt-T (right). We only show convolution/linear layers for visibility, which comprise 99.7% of total parameter count.

**Swin-T and ConvNeXt-T.**   Figure 7 shows the assigned learning rates on ImageNet-1k classification. Swin-T has a 2-2-6-2 block configuration, and the learning rates of Swin-T follow a similar pattern to ViT-S/16, except that the relative position layers in each block are assigned noticeably large learning rates from 3.99 to 7.77. ConvNeXt-T has a 3-3-9-3 block configuration, where each block consist of a depthwise separable convolution layer and then a multilayer perceptron block. Low learning rates are assigned to the depthwise seperable convolution layer and the second linear layer of the mlp block, while the first layer mlp block is assigned larger learning rates.

**GPT2 experimental details.**   We adopt a pytorch implementation from a publicly available codebase[1]. Weight decay of 0.1 for AdamW, 1.0 for Lion and 0.2 for Sophia is performed on the linear layers only. We use the same hyperparameters of AdamW for Adam-mini. For 9.89B token training a learning rate of 6e-4/2.4e-3, 1.2e-4/4.8e-4 and 1.5e-4/6e-4 is used for AdamW, Lion and Sophia respectively with a cosine learning rate schedule and 700 iteration warmup. When performing multiplicative residual downscaling we remove the $1/sqrt2 * B$ additional weight sampling in

---

[1]https://github.com/karpathy/llm.c

Table 6: ImageNet-1k hyperparameters

| Model | Dropout | Stoch Depth | Data Augmentation | Optimizer | Batch Size | lr | wd |
|-------|---------|-------------|-------------------|-----------|------------|----|----|
| | - | - | basic | SGD | 256 | 0.1 | 1e-4 |
| ResNet-50 | - | - | basic | AdamW | 1024 | 3e-3 | 0.1 |
| | - | - | strong | AdamW | 1024 | 3e-3 | 0.1 |
| | 0.1 | 0.1 | basic | SGD | 4096 | 0.8/1.6 | 1e-4 |
| ViT-S/16 | 0.1 | 0.1 | basic | AdamW | 4096 | 1e-2 | 0.1 |
| | - | - | strong | AdamW | 4096 | 1e-2 | 0.1 |
| Swin-T | - | 0.2 | strong | AdamW | 1024 | 1e-3 | 5e-2 |
| ConvNeXt-T | - | 0.1 | strong | AdamW | 4096 | 4e-4 | 5e-2 |

Table 7: ImageNet-1k real, v2 performance improvement due to layer-wise learning rate.

| Model | #Params | Optimizer | Data Augmentation | Epochs | Accuracy | |
|-------|---------|-----------|-------------------|--------|----------|--|
| | | | | | ReaL | V2 |
| | | SGD | basic | 90 | 83.63(+0.20) | 72.56(+0.33) |
| | | | | 200 | 84.18(+0.77) | 73.24(+0.58) |
| ResNet-50 | 25.56M | AdamW | basic | 200 | 82.94(+0.43) | 72.24(+0.71) |
| | | | strong | 300 | 85.24(+0.46) | 74.65(+0.47) |
| | | SGD | basic | 300 | 77.96(+2.59) | 65.99(+2.85) |
| ViT-S/16 | 22.05M | AdamW | basic | 300 | 81.44(-0.02) | 70.36(-0.24) |
| | | | strong | 300 | 84.19(+0.82) | 73.61(+1.28) |
| Swin-T | 28.29M | AdamW | strong | 300 | 85.13(-0.06) | 74.98(+0.31) |
| ConvNeXt-T | 28.59M | AdamW | strong | 300 | 85.61(+0.33) | 75.98(+0.18) |

the residual branches. The 124M model has vocabulary size of 50257, context size of 1024 and embedding dimension of 768, and is trained on bfloat16. A single batch consists of 524288 tokens which translates to a batch size of 512. We use a T of 100 for Algorithm 1.

Table 8: Train loss (↓) of small scale overfitting experiments with Adam-mini optimizer on 124M model next token prediction.

| Learning rate $\gamma$ | 7.5e-5 | 1.5e-4 | 3e-4 | 6e-4 | 1.2e-3 | 2.4e-3 |
|------------------------|--------|--------|------|------|--------|--------|
| Single | 3.800 | 2.755 | **1.903** | 3.350 | 2.845 | 7.315 |
| Layer-wise ($f_{in}$) | 4.362 | 3.409 | 2.183 | 1.136 | **0.838** | 2.093 |

Table 9: Train loss (↓) of small scale overfitting experiments with Sophia optimizer on 124M model next token prediction.

| Learning rate $\gamma$ | 1.875e-5 | 3.75e-5 | 7.5e-5 | 1.5e-4 | 3e-4 | 6e-4 |
|------------------------|----------|---------|--------|--------|------|------|
| Single | 3.071 | 2.012 | **0.989** | 2.722 | 7.232 | 7.494 |
| Layer-wise ($f_{in}$) | 3.577 | 2.541 | 1.410 | 0.742 | **0.538** | 3.525 |

