# OpenReview forum: "Initializing the Layer-wise Learning Rate"
_ICLR.cc/2025/Conference — Submitted to ICLR 2025_

### Official Review · Reviewer_CYkf · 2024-10-29

**Soundness:** 2
**Presentation:** 2
**Contribution:** 1
**Rating:** 5
**Confidence:** 5

**Summary:**

This paper proposes the use of a layerwise learning rate based on the gradient magnitude at initialization. The authors claim that this rescaling of learning rate at initialization takes into account the layerwise statistics at initialization without the network undergoing any training. Then they use the optimizer to test various setting of dataset and architectures to show the efficacy of the optimizer.

**Strengths:**

1) The optimizer is different from adaptive step size methods like ADAM which is per-parameter based and uses the curvature statistics at every update. Their method only requires computing the learning rate normalization at initialization.
2) Optimizers like ADAM has an additional computational and storage overhead due to updating the second order moment information whereas this work uses a static learning rate throughout.
3) Compared to SGD, their layer-wise learning rate method is easier to converge for complicated networks on large datasets.

**Weaknesses:**

The authors have tried to address the motivation clearly which I highlight in the strengths but I still think it may not be enough for the following reasons:

*1) Weak motivation*

Motivating the paper as simply mitigating ADAM's computation time is not enough. ADAM computes the preconditioner based on GGN (Gauss Newton) estimate of the Hessian at every iteration. Even ADAM is far from being a Newton-like update, it still calculates the curvature at the current iterate. The proposed method in this paper does not do that. The gradient information at initialization has no information of the curvature at any point in the loss landscape (except maybe at the initialization where curvature and gradient are correlated for eg in linear models). So, the fundamental motivation for the proposed algorithm seems to be missing.

*2) No substantial improvement*

The improvement on top of SGD or ADAM-W seems to be marginal and it is uncertain that whether it is solely because of the layerwise learning rate or the interaction of this layerwise lr with other tricks such as cosine scheudling or small batch size. It's hard to justify this algorithm unless studied in isolation without any other regularization tricks. Furthermore a 0.50% or 0.96% improvement is not considered substantial since there are various sharpness regularizers like SAM or even noise injection that outperforms their base optimizers (SGD or ADAM-W) by a siginficant margin. In this context the use of such layerwise learning rate is obsolete and unnecessary in the long run.

*3) Typos and unclear definitions*

Is there a typo in line 10 and 11 in Algorithm-1? seems like the sum cancels out?
I am not sure what the index t stands for in the algorithm. The authors say they use the gradient magnitude at initilaization as informaiton to scale the learning rate. If the authors are trying to denote the index t as sampling of a new minibatch, then they should use a t-depdent notation for $L$ and not just L. The current terminology is confusing. The authors use the phrase
"low learning rates are assigned to low-level layers and high learning rates to deeper, high-level layers" where it is not defined what low or high level layers mean.

*4) Lack of intuitive justification for method*

Even if there is minor improvement in certain tasks, there is no justification as to why there is an improvement. Is it possible to plot some training metrics such as sharpness and perform a correlation analysis with respect to layer-size and learning rate? Also, is it possible that scaling the learning rate based on the dimension of each layer can be used to achieve the same result?

**Questions:**

The authors can address the four weakness points above.

---

> ### Author Response · Authors · 2024-11-23
>
> We thank the reviewer for detailed and constructive feedback. We hope to sufficiently address your concerns and questions below.
>
> **W1** weak motivation
> > Motivating the paper as simply mitigating ADAM's computation time is not enough
> - We would like to clarify that our motivation is not necessarily pertained to replacing or mitigating a particular aspect of Adam, but to answer a more broader question of whether assigning non-adaptive static layer-wise learning rates can be a practical or effective stategy.
>
> > The gradient information at initialization has no information of the curvature at any point in the loss landscape. So, the fundamental motivation for the proposed algorithm seems to be missing.
> - We appreciate the reviewers concern in the proposed method does not have a strong theoretical connection, which may mean the presented results do not generalize.
> - However, in a situation where the connection between existing theory and empirical performance remains unclear, we believe approaching from different directions constitutes a valid motivation.
> - For example, [1] as mentioned by reviewer NtQ7 investigates the sign descent property of Adam and present evidence that counter the idea ADAM outperforms SGD on transformers due to a more robust estimate of the descent direction. Given evidence of the effectiveness of sign-descent optimizers [2], we consider exploring methods of that adjust the learning rate that at a layer-wise level in a static manner a reasonable motivation.
>
> **W2** No substantial improvement
> > The improvement on top of SGD or ADAM-W seems to be marginal and it is uncertain that whether it is solely because of the layerwise learning rate or the interaction of this layerwise lr with other tricks such as cosine scheudling or small batch size. It's hard to justify this algorithm unless studied in isolation without any other regularization tricks.
> - We agree the presented improvement on top of SGD or ADAM-W is not enough to ascertain that the layer-wise learning rate is beneficial given the many potential factors that could have affected the results.
> - The overfitting experiments on Table 2, 3 and in Table 8, 9 are settings designed to explicitly factor out the many potential tricks. It is a setting where we do not perform cosine scheduling and where we measure the train loss of the small dataset, such that factors that pertain to generalization or regularization would not affect the result. Arguably the small batch size trick remains, but in practical training a small batch size is likely to be relavent than performing a full gradient descent. The experiments performed on 4 optimizers (AdamW, Adam-mini, Lion, Sophia) demonstrate that the layer-wise method consistently improve train loss.
> - In particular, the additional experimenents with Sophia [3], which were performed during rebuttal phase, show that the layer-wise learning is also beneficial to a Hessian-based optimzier. It further suggests the existence of additional factors that can be leveraged to improve performance even when using computation-efficient second-order methods. Given the highly specialized architectures which modern neural network training is concerned with, how architecture interacts with training makes it a high priority for investigation.
>
> > various sharpness regularizers like SAM or even noise injection that outperforms their base optimizers (SGD or ADAM-W) by a significant margin. In this context the use of such layerwise learning rate is obsolete and unnecessary in the long run.
> - SAM and noise injection methods are methods that focus on improve generalization, at the cost of increased difficulty in training the original objective. Our experience with SAM is that while it improves validation accuracy there is a noticeable degradation, i.e. increase in train loss. Our understanding is that both methods that improve either convergability or generalization are significant in the long run.
>
> [1] Kunstner, Frederik, et al. "Noise Is Not the Main Factor Behind the Gap Between Sgd and Adam on Transformers, But Sign Descent Might Be." The Eleventh International Conference on Learning Representations.
>
> [2] Chen, Xiangning, et al. "Symbolic discovery of optimization algorithms." Advances in neural information processing systems 37 (2023).
>
> [3] Liu, Hong, et al. "Sophia: A Scalable Stochastic Second-order Optimizer for Language Model Pre-training." The Twelfth International Conference on Learning Representations.

---

> > ### Author Response · Authors · 2024-11-23
> >
> > **W4** Lack of intuitive justification for method
> > > there is no justification as to why there is an improvement.
> > - We argue initialization is a natural candidate to investigate architectural factors as it is a special case where it is not overshadowed by effects that arise during training, and has been investigated by the various weight initialization literature. The numerical values show that at initialization the layer-wise gradient differ in the scale of 1e4~1e5, and it would not be particularly surprising such a charactiristic is partially or implicitly carried over to prolonged training.
> >
> > > Is it possible that scaling the learning rate based on the dimension of each layer can be used to achieve the same result?
> >
> > - Scaling the layer-wise learning rate relative to $\sqrt{f_{in}}$ would achieve the most similar layer-wise pattern, in that the learning rates will increase as the layer is closer to the output for convolutional networks and that the word embedding layer would be assigned large learning rates compared to other layers in GPT-2. However, such method is not well defined for normalization scale and bias layers, and would assigns same learning rate to the q, k and v layers in transformers as they have the same input dimension. In comparison, the proposed method assigns larger values to q, k layers and lower v to value layers.
> >
> > **W3** Typos and unclear definitions
> > - We thank the reviewer for informing us of the typos and unclear definitions. The index t denotes sampling of a new minibatch, and have modified the manuscript to use a t-dependent notation for $L$. We used the phrase low level to indicate layers that are close to the input and high level layers to those close to the output, and added the clarifying phases to the manuscript.

---

> > > ### Comment · Reviewer_CYkf · 2024-11-25
> > > **Reviewer response**
> > >
> > > I thank the author for the detailed feedback. I appreciate that the authors were quite honest about their proposed method and I mostly agree with the response they provided. I do believe in deep learning, just like "Initializing the Layer-wise Learning Rate" there are numerous other small tricks (that lacks proper intuitive justificaiton) that can help improve the accuracy by some amount. So, if I were to ask myself "Would such work benefit the ML community in the long run?"..I would have to say my answer may not be so positive. But whatever the authors presented, was done clearly. So, based on the whole evaluation I update my score.

---

### Official Review · Reviewer_8G2V · 2024-11-02

**Soundness:** 2
**Presentation:** 3
**Contribution:** 2
**Rating:** 3
**Confidence:** 4

**Summary:**

The paper focuses on setting the static layer-wise learning rate. By conducting empirical experiments on large-scale datasets among different tasks, the authors find out that there is a trend of parameter-wise gradient during the training procedure. From this insight, the authors propose a simple method based on inverting gradient magnitude to set static learning rates per layer.

**Strengths:**

- The authors identify key points in their empirical observations.
- The authors combine empirical and theoretical findings from other papers to propose a simple yet effective method.
- The authors evaluate the proposed method with different model architectures.

**Weaknesses:**

- The main weakness of this paper comes from the empirical study. In detail, the empirical study is potentially sensitive to the weight initialization mechanism. Therefore the obtained results are not enough to conclude the main findings of the paper.
- This method requires several prior training steps to approximate the gradient trajectory and then calculate the static learning rate, which may increase the total training time.
- Some statements are not theoretically or empirically proven.
- There are some points that the empirical analysis is not aligned with the justification of the proposed method.

**Questions:**

1. Did the authors conduct empirical studies with different weight initialization? Or can the authors provide any theoretical study to prove the finding observation?
2. Did the authors conduct an ablation test on the combination of methods with different kinds of weight initialization mechanisms?
3. Can the authors explain why the layer-wise scheme is used while the previous weigh initialization technique was not considered?
4. Can the authors provide suggestions for the number of prior training steps needed for approximating gradient trajectory theoretically?
5. The author said that “For ResNet-50, using a T of 1 instead of 5004 results in a max 11.02%, average 2.66% deviation in the assigned layer-wise learning rates, which would not significantly impact the network performance compared to other random factors.”. However, in Figure 1, the empirical study for ResNet50 is conducted for 90 epochs, which is much longer than T = 1. Can the author provide more information on this unalignment?
6. The author said, “A setting where a smaller subset of the dataset is repeatedly iterated over can be an efficient method to check if it can learn the characteristics implicitly shared across many minibatch, at the drawback of not explicitly verifying it on the large, diverse full dataset”. Did the author experiment with large batch size and with an optimizer for large batch training (Lars [R1], Lamb [R2], etc)?

References:
- [R1]: Boris Ginsburg, Igor Gitman, Yang You. Large Batch Training of Convolutional Networks with Layer-wise Adaptive Rate Scaling. ICLR 2018
- [R2]: Yang You, Jing Li, Sashank Reddi, Jonathan Hseu, Sanjiv Kumar, Srinadh Bhojanapalli, Xiaodan Song, James Demmel, Kurt Keutzer, Cho-Jui Hsieh. Large Batch Optimization for Deep Learning: Training BERT in 76 minutes. ICLR 2020.

---

> ### Author Response · Authors · 2024-11-23
>
> We thank the reviewer for detailed and constructive feedback. We hope to sufficiently address your concerns and questions below.
>
> **Q1, Q2** and **Q3**: Regarding the combinations with different weight initializations with the proposed method
>
> - Many weight initialization methods previously proposed would have been evaluated on settings where a single learning rate is used. This means that the performance of such methods could have tuned to be favorable at such settings. As such, when performing ablation studies in Table 2, we focused on the most basic initialization schemes that were initially proposed, that is, the activation preserving and gradient preserving fan-in and fan-out initialization that samples weights from $N(0, 1/f_{in})$ and $N(0, 1/f_{out})$
> - We further provide results from additional initialization experiments below, where we experiment with downweighting the residual branches $h(x) = x + \beta f(x)$. We would like to note the $\beta$ scaling is incorporated as a multiplicative scalar rather than just adjusting the variance from which the residual branch weight was sampled from.
> - As it is a initialization scheme concerned with scalability with depth, we provide results on the 774M GPT-2 model that has triple the depth (36) compared to 124M GPT-2 model (12). Due to computational constraints, we provide the train loss (↓)  results from small scale overfitting experiments when trained with Lion.
>
> | Learning rate | 3.75e-5 | 7.5e-5 | 1.5e-5 | 3e-5 | 6e-5 | 1.2e-4 | 2.4e-4 |
> | --- | --- | --- | --- | --- | --- | --- | --- |
> Single | 2.647 | 0.840 | 0.316 | 0.092 | 0.301 | 4.952 | -  |
> Layer-wise | 2.528 | 0.452 | 0.105 | 0.241 | 0.488 | 7.174 | - |
> Single-$\beta$ | - | 2.513 | 0.367 | 0.059 | 0.156 | 6.403 | 7.537 |
> Layer-wise-$\beta$ | - | 0.748 | 0.034 | **0.019** | 0.026 | 0.056 | 7.492 |
>
> - Results show the method further improves convergability on top of the additional $\beta$ scaling, demonstrating that the method can further benefit from advancements in initialization schemes. We updated the manuscript to include the result in Table 5 and the accompanying description.
>
> **Q4**: number of prior training steps needed to approximating gradient trajectory
> - The layer-wise gradient is measured as the average per parameter gradient magnitude of a layer at initialization. As the network is yet to have trained on a single minibatch, there is little deviation in the calculated values with respect to the actual iterations taken to measure to gradient. We recommend using a T of 100 and is the value used in the nlp experiments as it would still make the estimation more accurate and would take neglibible time compared to the entire training duration under most settings.
>
> **Q5**: connection between using T=1 and the empirical study for ResNet50 in Figure 1
>
> - The empirical study in Figure 1 is tracking and visualizing the step size modifier of various optimizers (AdamW, Lars and Lamb) when compared to SGD and AdamW. We show the per parameter values that is averaged across a layer and a single epoch. The value of T used refers to the number of minibatches used to measure the gradient at initalization for the proposed method, and is not performed in the empirical study in Figure 1. We are unsure of the question and would be glad to answer upon further clarification.
>
> **Q6**: experiments with optimizers for large batch training (Lars, Lamb)
> - Lars and Lamb are optimizers designed for large batch training, and while are also called "layer-wise learning rate" their motivation and methodology are different making them rather independent from our method.
> - We have added figures for Lamb in the empirical study of Figure 1, which shows that for ResNet50, both Lars and Lamb show little layer-wise patterns. In particular, the common implementation of Lamb assigns very low values to the conv/linear layers while batchnorm layers are not modified, creating a large discrepancy in the assigned learning rates. On the other hand, our method gracefully handles the batchnorm and layernorm layers which are naturally assigned similar values to adjacent layers. We consider investigating the effect of assigning large learning rates to batchnorm or layernorm layers beyond the scope of our work.

---

> > ### Comment · Reviewer_8G2V · 2024-11-26
> > **Reviewer response**
> >
> > I thank the authors for a significant effort to answer all the questions. However, I will not change my score. I have the same opinion as the Reviewer CYkf that the paper's contribution is incremental and has no theoretical justification. Otherwise, the contribution does not have a high impact on the field, instead, the authors should find a problem and dive into finding new angles which might bring big benefits.

---

### Official Review · Reviewer_NtQ7 · 2024-11-04

**Soundness:** 2
**Presentation:** 3
**Contribution:** 2
**Rating:** 3
**Confidence:** 3

**Summary:**

This paper addresses a vital problem of learning rate assignment in the field of optimization. The authors propose a simple yet effective method for assigning layer-wise learning rates based on variations in gradient magnitude. Experiments conducted across various model architectures demonstrate that their proposed approach significantly enhances training stability.

**Strengths:**

The non-adaptive layer-wise learning rate assignment method proposed in this paper is straightforward to implement, and experimental results demonstrate its effectiveness in stabilizing the training process.

**Weaknesses:**

- The novelty of the proposed method is somewhat limited. The authors in [1] have highlighted that layer heterogeneity impacts learning rate assignment and have introduced a similar approach in [2]. They validate their hypothesis by analyzing the block-wise Hessian spectra of the models. However, the evidence provided in this paper to support the effectiveness of the proposed methods is primarily intuitive.
- The paper lacks a theoretical justification for the presented method, particularly in terms of convergence analysis. Additionally, since the authors assert that the proposed methods can enhance model generalization, it would be beneficial for them to conduct a basic theoretical analysis to support this claim, such as deriving improved generalization error bounds.
- The experimental section is somewhat limited. The authors could enhance their evaluation by comparing their proposed method with more cutting-edge optimizers, such as Adam-Mini [2] and Sophia [3], to assess the performance benefits and overhead associated with their approach.

**Questions:**

- The Lion optimizer used in the experimental part addresses the impact of gradient magnitude through the use of the sign operation. Given this, what is the rationale behind the proposed method's effectiveness in conjunction with this optimizer? Could the authors offer some theoretical justification for this phenomena?
- What is the fundamental novelty and distinction of the proposed methods' underlying motivation or theoretical foundations in comparison to previous works, such as [1], which emphasizes block-wise heterogeneous Hessian spectra, and [4], which highlights the significance of gradient magnitude in the Adam optimization algorithm?

[1] Zhang, Y., Chen, C., Ding, T., Li, Z., Sun, R., & Luo, Z. Q. (2024). Why transformers need adam: A hessian perspective. arXiv preprint arXiv:2402.16788.

[2] Zhang, Y., Chen, C., Li, Z., Ding, T., Wu, C., Ye, Y., ... & Sun, R. (2024). Adam-mini: Use fewer learning rates to gain more. arXiv preprint arXiv:2406.16793.

[3] Liu, H., Li, Z., Hall, D., Liang, P., & Ma, T. (2023). Sophia: A scalable stochastic second-order optimizer for language model pre-training. arXiv preprint arXiv:2305.14342.

[4] Kunstner, F., Chen, J., Lavington, J. W., & Schmidt, M. (2023). Noise is not the main factor behind the gap between sgd and adam on transformers, but sign descent might be. arXiv preprint arXiv:2304.13960.

---

> ### Author Response · Authors · 2024-11-23
>
> We thank the reviewer for detailed and constructive feedback. We hope to sufficiently address your concerns and questions below.
>
> **W1** and **Q4**: Novelty compared to [1] and [2]
>
> - The proposed method's layer-wise learning rate adjust the "macro" level step size such that the per-layer step sizes become **significantly different from sign descent**. As the assigned learning rates are non-adaptive values, it affects the entire training duration and is an additional multiplicative factor on top of the optimizer specific step size adjustments, which we will refer to as "micro" level step size adjustment.
>
> - In contrast, the layer-wise learning rate of [2] is effectively using layer-wise average $1/\sqrt{v}$ instead of per-parameter $1/\sqrt{v}$, and mentions its similarity to Adam as a major characteristic. It is effectively a replacement/approximation to the "micro" level step size adjustment already performed by Adam.
>
> **W1**: evidence provided in this paper to support the effectiveness of the proposed methods is primarily intuitive
>
> - We provide extensive experiments that the "macro" level adjustment is effective over varying types of architectures and optimizers, that show strong empirical evidence the effectiveness is not limited to a particular optimizer or architecture.
>
> **W2**: lacks a theoretical justification for the presented method, **Q1**: rationale behind the proposed method's effectiveness in conjunction with Lion optimizer, **Q4**: novelty and distinction from [4]
>
> - Due to analytical convenience, theory behind initialization has been mostly focus at initialization to investigate architectural factors as it is a special case where it is not overshadowed by effects that arise during training. We demonstrate that initialization characteristics can be leveraged to improve training stability through the assignment of layer-wise learning rates at initialization. While theoretical justification would be beneficial, we believe empirical evidence of the effectiveness of the techniques is also valuable, and that initialization characteristics can carry on to prolonged training has also been demonstrated in [5].
>
> - The effectiveness in conjunction with Lion further demonstrates that the "macro" level adjustment according to initialization characterisctics accounts for the architectural factors that is difficult to measure from in-training time gradients, and by extension makes it different the "micro" level step size adjustments. It shows the benefit of the additional multiplicative factor performed in conjunction with optimizers that have sign-descent properties (Lion, Adam), and aligns with the claims of [4] that having uniform changes despite large changes in (in-training) time gradients is a contribution to the success of Adam.
>
> **W2**: The authors assert that the proposed methods can enhance model generalization
>
> - The main argument is that the proposed method can improve convergability, which we demonstrate by showing that layer-wise methods significantly improve train loss when overfitting on small dataset when trained with AdamW and Lion in Table 2 and 3. We performed additional experiments that further demonstrate improved convergability with different optimizers (Adam-mini, Sohpia) in Table 8, 9 in the appendix and when scaling model depth in Table 5.
>
> **W3**: The authors could enhance their evaluation by comparing their proposed method with more cutting-edge optimizers, such as Adam-Mini [2] and Sophia [3]
>
> - Per reviewer's request, we provide additional experiments on Adam-mini and Sophia. The table below shows the lowest train loss when sweeping learning rates in small scale overfitting experiments and the validation loss when trained for 9.89B tokens.
>
> - The layer-wise scheme improves train loss for all optimizers in small scale experiments, demonstrating the different effect of assigning static learning rates compared to Adam-mini. Interestingly, in 9.89B training the validation loss is slightly behind for both Adam-mini and AdamW, demonstrating that Adam-mini's layer-wise learning scheme is effectively approximating Adam. In fact, our results suggest that there is also a slight performance degradation when approximating AdamW through Adam-mini. The method also results in a noticable improvement for Sophia in 9.89B training.
>
> |**Optimizer** | AdamW | AdamW-Lw | Adam-mini | Adam-mini-lw | Lion | Lion-Lw | Sophia | Sophia-Lw |
> | --- | --- | --- | --- | --- | --- | --- | --- | --- |
> | small scale train loss (↓) | 1.430 | **0.560** | 1.903 | **0.838** | 0.704 | **0.179** | 0.989 | **0.538** |
> | 9.89B validation loss (↓) | **3.3344** | 3.3540 | **3.3707** | 3.3844 | 3.2888 | **3.2851** | 3.3523 | **3.3153** |
>
> [5] He, Bobby, et al. "Understanding and Minimising Outlier Features in Neural Network Training." arXiv preprint arXiv:2405.19279 (2024)

---

> > ### Comment · Reviewer_NtQ7 · 2024-11-29
> >
> > Thank you for your response. While I appreciate the authors' efforts to demonstrate that initialization characteristics can be leveraged to improve training stability, I noticed that they have not provided a rigorours mathematical definition of training stability or conducted the corresponding theoretical analysis to substantiate how their methods achieve this. Therefore, I would like to maintain my score.

---

### Author Response · Authors · 2024-11-23
**Additional experimental results**

We thank the reviewer for their time and effort in reviewing our paper. Below we summarize the additional experimental results added to the manuscript.

Figure 1 has been modified to also include step size multiplier due to Lamb compared to AdamW, which demonstrates the differences of Lamb's weight_norm/grad_norm scaling compared to the the proposed layer-wise scheme and also how Lamb assigns relatively large learning rates to the batchnorm layers.

Results on Adam-mini and Sophia in small scale overfitting experiments and 9.89B token training have been added in Table 8, 9 and in Table 4.

Experiments that verify scalability with depth have been added in Table 5. We experiment on 774M GPT2 model, which has 36 attention blocks compared to 12 of the 124M model. Results demonstrate the scalablity of the method when combined with $1/\sqrt{depth}$ residual branch scaling, strengthening the connection of the method with existing initialization schemes.

---

### Meta-Review · Area_Chair_7SQS · 2024-12-15

**Metareview:**

The paper presents an approach to initializing layer-wise learning rate and empirically demonstrates the effectiveness of the proposed approach. The work presents the ideas clearly and the proposed ideas are easy to implement.

The primary concerns from several of the reviewers are that the method has no theoretical motivation or justification, the empirical gains are not substantial, and, as pointed out by some, the gains will likely be overshadowed by existing approaches such as SAM or noise injection. The reviewers mostly engaged with the authors during the discussion phase, but the concerns persisted.

**Additional Comments On Reviewer Discussion:**

The reviewers engaged with the authors and responded briefly but clearly to the author responses. One reviewer increased their score.

---

### Decision · Program_Chairs · 2025-01-22

Reject